# No increased risk of cancer associated with metal-on-metal or ceramic-on-ceramic procedures compared to other bearing surfaces in patients with total hip arthroplasty: A nationwide linked registry cohort analysis of 167,837 patients

Nicole L. Pratt[1]*, Flavia M. Cicuttini[2], Yuanyuan Wang[2], Stephen E. Graves[3,4]

1 Quality Use of Medicines and Pharmacy Research Centre, Clinical and Health Sciences, University of South Australia, Adelaide, South Australia, Australia, 2 Department of Epidemiology and Preventive Medicine, School of Public Health and Preventive Medicine, Monash University, Melbourne, Victoria, Australia, 3 Australian Orthopaedic Association National Joint Replacement Registry, South Australian Health and Medical Research Institute, Adelaide, South Australia, Australia, 4 Clinical and Health Sciences, University of South Australia, Adelaide, South Australia, Australia

* Nicole.Pratt@unisa.edu.au

**Data Availability Statement:** Data cannot be shared publicly due to restrictions imposed by the

## Abstract

### Objectives

Studies have identified increased cancer risk among patients undergoing total hip arthroplasty (THA) compared to the general population. However, evidence of all-cause and site-specific cancer risk associated with different bearing surfaces has varied, with previous studies having short latency periods with respect to use of modern Metal-on-Metal (MoM) bearings. Using the Australian Orthopaedic Association National Joint Replacement Registry (AOANJRR) linked to Australasian Association of Cancer Registries data, our aim was to evaluate risk of all-cause and site-specific cancer according to bearing surfaces in patients undergoing THA for osteoarthritis and whether risk increased with MoM bearings.

### Methods

Standardized incidence ratios (SIRs) and 95% confidence intervals (CIs) were calculated by comparing number of observed cancer cases to expected number based on incidence rate in the Australian population. All-cause and site-specific cancer rates were calculated for all conventional stemmed THA (csTHA) and resurfacing THA (rsTHA) procedures performed for osteoarthritis. Cox proportional hazards models were used to compare cancer rates for MoM, ceramic-on-ceramic (CoC) and resurfacing procedures with a comparison group comprising metal-on-polyethylene (MoP) or ceramic-on-polyethylene (CoP) procedures.

approving ethics committee and due to the privacy and confidentiality provisions imposed by the data custodians. The research data for this study was created by the Australian Institute of Health and Welfare (AIHW) from pre-existing datasets (Australian Orthopaedic Association National Joint Replacement Registry (AOANJRR) and the Australian Cancer Database), following receipt of data custodian permissions and specific ethical approvals. As data contain sensitive patient information and to ensure privacy and confidentiality of participants data are maintained, data are unable to be shared publicly with restrictions (on data scope, format and personnel access) imposed by the approving Ethics Committees and dataset custodians. The data may potentially be made available to other researchers subject to Ethics Committee approvals and agreement of all relevant data custodians in all states and territories. Interested parties should first contact the AIHW Australian Cancer Database at cancer@aihw.gov.au Further information about data access and the linkage process is available from, https://www.aihw.gov.au/about-our-data/our-data-collections/australian-cancer-database, https://www.aihw.gov.au/our-services/data-linkage and for the AOANJRR, https://aoanjrr.sahmri.com/aoanjrr-data-linkage or via, admin@aoanjrr.org.au.

**Funding:** The following authors (YW, SEG, FMC & NLP) received funding from the Australian Orthopaedic Association Research Foundation (https://aoa.org.au/research/research-foundationM/research-foundation) for the work. Authors SEG and NLP also received funding from the Australian National Health & Medical Research Council Project Grant APP1148106 which supported the work (https://www.nhmrc.gov.au/funding/find-funding/project-grants). The funders played no role in the study design, data collection and analysis, decision to publish, or preparation of the manuscript.

**Competing interests:** The authors have declared that no competing interests exist.

## Results

There were 156,516 patients with csTHA procedures and 11,321 with rsTHA procedures for osteoarthritis performed between 1999 and 2012. Incidence of all-cause cancer was significantly higher for csTHA (SIR 1.24, 95% CI 1.22–1.26) and rsTHA (SIR 1.74, 95% CI 1.39–2.04) compared to the Australian population. For csTHA, there was no significant difference in all-site cancer rates for MoM (Hazard Ratio (HR) 1.01, 95%CI 0.96–1.07) or CoC (HR 0.98, 95%CI 0.94–1.02) compared to MoP and CoP bearings. Significantly increased risk of melanoma, non-Hodgkins lymphoma, myeloma, leukaemia, prostate, colon, bladder and kidney cancer was found for csTHA and, prostate cancer, melanoma for rsTHA procedures when compared to the Australian population, although risk was not significantly different across bearing surfaces.

## Conclusions

csTHA and rsTHA procedures were associated with increased cancer incidence compared to the Australian population. However, no excess risk was observed for MoM or CoC procedures compared to other bearing surfaces.

## Introduction

Total hip arthroplasty (THA) is a highly efficacious and cost-effective treatment for symptomatic advanced joint disease [1–4]. Over the years, different materials have been used in joint arthroplasty prostheses and a number of different bearing surfaces have been developed. The most commonly used bearings are metal heads on a polyethelene acetabular component (MoP). Hip arthroplasties using a metal head on a metal acetabular component (MoM) gained popularity as this combination was believed to confer benefit due to decreased wear and osteolysis compared with MoP hip replacements [5]. The design of the modern MoM implants also allowed for the use of larger femoral head sizes, which could further reduce risk of revision due to dislocation [5]. However, in practice, MoM total hip procedures experienced a higher rate of revision compared with MoP implants, with concerns also raised about metal ion toxicity resulting from wear and corrosion of the bearing surfaces and the subsequent increase in serum metal ion levels [6, 7]. Patients with MoM total joint replacement have elevated concentrations of cobalt, chromium, and titanium in serum and urine [6–8], with even higher concentrations identified following prosthesis failure [8]. Patients with MoM hip replacement are exposed to significant amounts of metals such as cobalt and chromium with serum levels of more than 1000 times that of normal levels found in some patients [9] and generally, higher compared to THA patients receiving MoP [10], ceramic-on-polyethylene (CoP) [11] or ceramic-on-ceramic (CoC) [12] bearing surfaces.

It has been postulated that wear debris from worn prostheses in human tissue cultures damage chromosomes in a dose-dependent manner, specific to the type of metal [13]. Significant amounts of metal ions and particles have been found in human tissues surrounding orthopaedic implants and tissues at remote sites including lymph nodes, bone marrow, liver, and spleen through processes of corrosion and mechanical wear of the prosthesis [14]. The potential toxicity of metal wear debris associated with MoM hip replacements induces potential harmful effects on immunity, reproduction, the kidney, developmental toxicity and nervous system [15]. Although it is biologically plausible for the materials used in MoM hip replacement to

induce malignant degeneration, the relationship has not been well demonstrated. Many studies have examined the differential risk of cancer between different bearing surfaces, results have varied [16–33].

Therefore, using an Australian joint replacement registry database linked with cancer registry data, our aim was to evaluate the risk of all-cause and site-specific cancer according to bearing surfaces used in patients undergoing THA, either conventional stemmed THA (csTHA) or resurfacing THA (rsTHA) for osteoarthritis and whether risk was increased with MoM bearings.

## Methods

### Study design, setting and data sources

A retrospective registry cohort study was performed using linked data from the Australian Orthopaedic Association National Joint Replacement Registry (AOANJRR) and the Australasian Association of Cancer Registries (AACR).

The AOANJRR has recorded all hip and knee joint replacement procedures from 1 September 1999, becoming completely national by mid-2002. The AOANJRR collects a defined minimum dataset at time of surgery that comprises patient characteristics including age and sex, surgical indication (diagnosis reason for surgery e.g. osteoarthritis, tumour, fractured neck of femur), prosthesis type and features, method of prosthesis fixation and surgical technique. Although data collection for the Registry is voluntary, it receives cooperation from all hospitals undertaking joint arthroplasty surgery [34]. The AOANJRR validates its data by using both internal systems and external data sources. Each year, the AOANJRR dataset is linked with death registration data submitted by Australia's eight jurisdictions to the National Death Index (NDI). The NDI database is maintained by Australian Institute of Health and Welfare (AIHW) and contains records of all deaths occurring in Australia since 1980. As well as fact of death, date of death and demographic information, other variables recorded in the NDI include underlying causes of death and other cause-of-death information. Cause of death information is coded by the Australian Bureau of Statistics using the International Statistical Classification of Diseases and Related Health Problems Tenth revision (ICD-10) with non-specific causes revised up to two years following registration of death [35]. Deaths identified after the first primary total hip arthroplasty operation were included in the analysis to censor subjects.

The Australasian Association of Cancer Registries consists of data from Australia's eight state and territory cancer registries, the New Zealand Cancer Registry and the National Cancer Statistics Clearing House. Data on primary, malignant cancers diagnosed in Australia since 1982 are provided by the individual cancer registries to the AACR and compiled as the Australian Cancer Database (ACD) by the Australian Institute of Health and Welfare.

Probabilistic data linkage was used to link patient-level AOANJRR joint procedural data, first to the Medicare Enrolment File (containing Australia's universal healthcare identifier) and then to the Australian Cancer Database, using first given name, surname, date of birth, and gender. All AOANJRR patients with a THA performed in Australia between September 1, 1999 (commencement of AOANJRR data collection), and December 31, 2013 (most recent and complete cancer registration data available for all Australian states from the Australian Cancer Database at time of linkage) were linked to ACD patients. Records were successfully matched to the Medicare file for 97.8% of patients with an AOANJRR-recorded hip arthroplasty procedure, with 20.1% of the Medicare-AOANJRR matched patients linked to the Australian Cancer Database.

## Study population

Since each total hip arthroplasty is entered in the AOANJRR as a separate operation and cancer risks apply to patients rather than joints, all patients were considered to be at risk after their first recorded primary total hip arthroplasty operation identified in the registry performed for osteoarthritis, (specified in surgical indication). In Australia over 88% of csTHA procedures are performed for osteoarthritis and over 95% of resurfacing procedures are performed for osteoarthritis [36]. Procedures were included if they occurred before December 31 2012, this was to allow for at least one year of follow-up as cancer data were only available to the end of 2013 (Fig 1). Follow-up for cancer and calculation of individual person-time was determined from the date of first recorded total hip arthroplasty until either cancer diagnosis, death, or December 31, 2013, whichever occurred first. Person-time accrued and cancer events observed during the first year of follow-up were not included in the analysis, since cancer cases diagnosed during this time period are assumed to be coincidental, prone to selection and surveillance bias, and unlikely to be causally related to the implants. Only the first cancer after

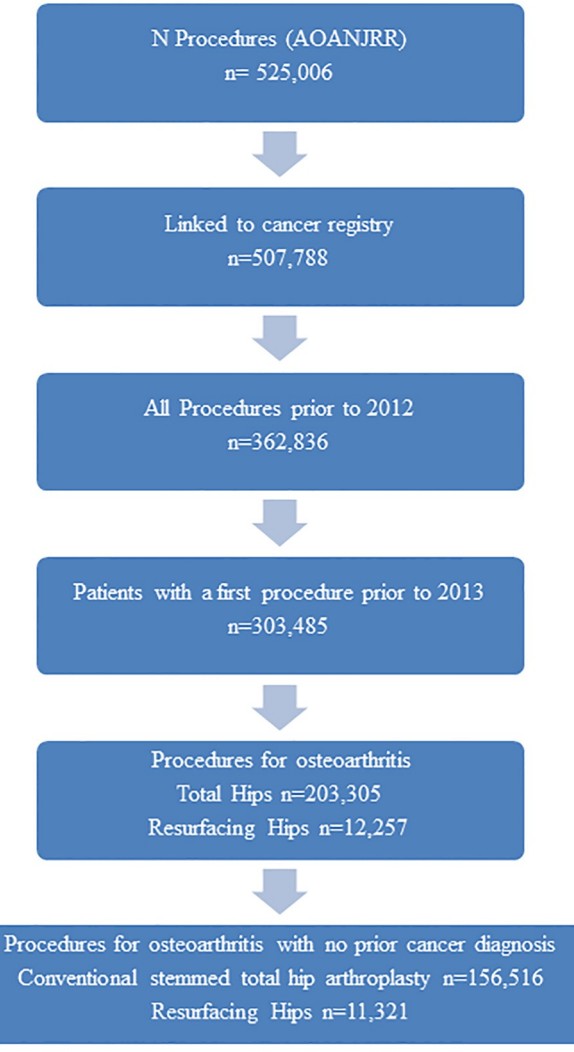

**Fig 1. Patient population and attrition.**

primary procedure was considered and those with any cancer diagnosed prior to their joint replacement procedure were excluded from this analysis.

## Statistical analysis

Descriptive statistics were used to summarise the demographics and follow-up time for the two classes of hip arthroplasty, csTHA and rsTHA and three bearing surfaces (MoM, CoC, MoP/CoC combined).

Standardized incidence ratios (SIRs) with 95% confidence intervals (CIs) were calculated for all cancer types and for site-specific cancer S1 Table. The number of expected cancer cases by sex, 10-year age group and calendar year was calculated by multiplying the number of person-years in each category observed in the registry by the corresponding incidence of cancer in Australia as a whole with respect to the same sex, age group, and calendar year. Australian cancer incidence data were obtained from the Australian Institute of Health and Welfare website [37]. SIRs were then calculated by dividing the observed numbers of cases by the expected cases. The standardisation takes into account the possible confounding due to differences in gender and age distribution between the joint replacement cohort and the general population, and the variations in cancer incidence over calendar years.

For all cancers overall and for site-specific cancers we stratified by bearing surface. MoM replacements, CoC and resurfacing procedures were compared to other bearing surfaces including MoP and CoP. Cox proportional hazards regression, adjusting for age and gender, was used to compare the cancer incidence rates between the bearing surface groups.

The SAS statistical program (version 9.4) was used for all analyses.

## Ethics review statement

Ethics approval for the study was granted by the following committees, Monash University Human Research Ethics Committee (HREC 16962), Cancer Council Victoria HREC (HREC 1402), ACT Health HREC (ETH3.14.055), South Australian Department of Health & Wellbeing HREC (HREC/14/SAH/32; 2021/HRE00328), Tasmanian Health & Medical HREC (H0014022), Australian Institute of Health & Welfare (AIHW) Ethics Committee (EO2014/2/84), NSW Population & Health Services Research Ethics Committee (HREC/14/CIPHS/27), WA Department of Health HREC (2014/27). Ethics approval granted by these committees included approval for a consent waiver with data analysed anonymously.

## Results

There were 156,516 csTHA patients included with median age 68 years and 5 years follow-up and 11,321 rsTHA patients with median age 53 years and 8 years follow-up. There were differences in the population demographics of patients according to the bearing surface used Table 1. The comparison group (MoP and CoP) were older and had a higher proportion of women than CoC and MoM procedures, while patients with rsTHA were younger and a higher proportion were men compared to patients with a csTHA.

All-cause cancer risk for csTHA procedures was significantly higher than the Australian population (SIR 1.24, 95% Confidence Interval (CI) 1.22–1.26) Table 2. Site-specific cancer rates were significantly higher than the general population for prostate cancer (SIR 1.46, 95% CI 1.41–1.51), melanoma (SIR 1.53, 95% CI 1.46–1.61), non-Hodgkin's lymphoma (SIR 1.25, 95% CI 1.14–1.35), myeloma (SIR 1.60, 95% CI 1.41–1.79), leukaemia (SIR 1.21, 95% CI 1.10–1.33), colon (SIR 1.24, 95% CI 1.18–1.30), bladder (SIR 1.15, 95% CI 1.03–1.27) and kidney (SIR 1.22, 95% CI 1.09–1.35) Table 2. Similar results were observed for individual bearing surfaces for the csTHA procedures. For rsTHA procedures all-cause cancer risk was significantly

**Table 1. Demographic characteristics of patients undergoing conventional stemmed THA or resurfacing THA by bearing surface.**

| | Patients (N) | Age at primary arthroplasty, years (median (IQR)) | Male (N (%)) | Person years | Follow-up, years (median (IQR)) |
|---|---|---|---|---|---|
| Conventional stemmed total hip replacement procedures | 156,516 | 68 (60–75) | 70569 (45%) | 920353 | 5 (3–8) |
| **Bearing Surface*** | | | | | |
| *MoP CoP* | 105,485 | 71 (64–77) | 44849 (42.5%) | 619112 | 5 (3–9) |
| *CoC* | 36,481 | 63 (56–69) | 17854 (51.1%) | 203424 | 5 (3–8) |
| *MoM* | 14,221 | 63 (56–70) | 7988 (56%) | 96086 | 7 (5–8) |
| Resurfacing Hip | 11,321 | 53 (47–59) | 8787 (78%) | 83972 | 8 (5–10) |

*Note that bearing surface could not be classified in 329 conventional stemmed total hip arthroplasty procedures

Abbreviations: MoP = Metal-on-polyethylene, CoP = Ceramic-on-polyethylene, MoM = Metal-on-Metal, CoC = Ceramic-on-Ceramic, IQR = Interquartile range.

higher than the Australian population (SIR 1.42, 95% CI 1.32–1.51) Table 2. The individual cancer sub-types where risk for the rsTHA population was significantly higher than the general population were prostate cancer (SIR 2.30, 95% CI 2.05–2.55) and melanoma (SIR 1.72, 95% CI 1.39–2.04) Table 3. Lower risks were observed for lung cancer (SIR 0.68, 95% CI 0.44–0.91).

**Table 2. Standardised incidence ratios for primary cancer diagnoses for patients receiving a conventional stemmed Total Hip or Resurfacing Hip arthroplasty for osteoarthritis between 1999 and 2012.** Expected numbers of cancer cases are approximated from the incidence rate of cancer for the general Australian population.

| | | | | Conventional Stemmed THA by bearing surface | | | | | | | | | | | |
|---|---|---|---|---|---|---|---|---|---|---|---|---|---|---|---|
| | Conventional Stemmed THA | | | MoP[1] CoP[2] | | | MoM[3] | | | CoC[4] | | | Resurfacing THA | | |
| | Obs | Exp | SIR (95% CI) | Obs | Exp | SIR (95% CI) | Obs | Exp | SIR (95% CI) | Obs | Exp | SIR (95% CI) | Obs | Exp | SIR (95% CI) |
| **All Cause** | 15565 | 12541 | 1.24 (1.22–1.26) | 11161 | 9154 | 1.22 (1.20–1.24) | 1502 | 1137 | 1.32 (1.25–1.39) | 2878 | 2228 | 1.29 (1.24–1.34) | 815 | 576 | 1.42 (1.32–1.51) |
| **Prostate** | 3165 | 2165 | 1.46 (1.41–1.51) | 2081 | 1496 | 1.39 (1.33–1.45) | 390 | 240 | 1.63 (1.47–1.79) | 684 | 426 | 1.61 (1.49–1.73) | 325 | 141 | 2.30 (2.05–2.55) |
| **Melanoma** | 1535 | 1001 | 1.53 (1.46–1.61) | 1044 | 711 | 1.47 (1.38–1.56) | 169 | 97 | 1.74 (1.47–2.00) | 320 | 192 | 1.67 (1.49–1.85) | 108 | 63 | 1.72 (1.40–2.04) |
| **Non-Hodgkins** | 575 | 462 | 1.25 (1.14–1.35) | 420 | 340 | 1.23 (1.12–1.35) | 58 | 40 | 1.43 (1.06–1.80) | 97 | 80 | 1.21 (0.97–1.45) | 31 | 21 | 1.45 (0.94–1.96) |
| **Myeloma** | 275 | 172 | 1.60 (1.41–1.79) | 209 | 129 | 1.62 (1.40–1.84) | 23 | 14 | 1.61 (0.95–2.27) | 43 | 28 | 1.53 (1.08–1.99) | 7 | 6 | 1.12 (0.29–1.95) |
| **Leukaemia** | 417 | 344 | 1.21 (1.10–1.33) | 306 | 257 | 1.19 (1.06–1.32) | 41 | 29 | 1.40 (0.97–1.83) | 69 | 57 | 1.22 (0.93–1.50) | 18 | 14 | 1.26 (0.68–1.85) |
| **Lung** | 1275 | 1343 | 0.95 (0.90–1.00) | 979 | 1000 | 0.98 (0.92–1.04) | 92 | 116 | 0.79 (0.63–0.95) | 202 | 225 | 0.90 (0.77–1.02) | 33 | 49 | 0.68 (0.44–0.91) |
| **Colon** | 1564 | 1258 | 1.24 (1.18–1.30) | 1188 | 953 | 1.25 (1.18–1.32) | 143 | 103 | 1.39 (1.17–1.62) | 233 | 200 | 1.16 (1.01–1.31) | 44 | 41 | 1.08 (0.76–1.40) |
| **Bladder** | 353 | 308 | 1.15 (1.03–1.27) | 274 | 236 | 1.16 (1.02–1.30) | 27 | 25 | 1.09 (0.68–1.50) | 52 | 47 | 1.11 (0.81–1.41) | 7 | 10 | 0.69 (0.18–1.20) |
| **Kidney** | 341 | 279 | 1.22 (1.09–1.35) | 233 | 200 | 1.17 (1.02–1.31) | 34 | 27 | 1.27 (0.84–1.69) | 73 | 52 | 1.41 (1.09–1.73) | 22 | 16 | 1.36 (0.79–1.92) |

Abbreviations:

[1] MoP = Metal-on-polyethylene,

[2] CoP = Ceramic-on-polyethylene,

[3] MoM = Metal-on-Metal,

[4] CoC = Ceramic-on-Ceramic, SIR = Standardized incidence ratios, CI = Confidence interval, THA = Total hip arthroplasty.

**Table 3. Risk of cancer incidence for all-cause and site-specific cancers by bearing surface.**

| | MoP[1] CoP[2] | MoM[3] | CoC[4] | Resurfacing |
|---|---|---|---|---|
| | | HR (95% CI) | HR (95% CI) | HR (95% CI) |
| **All Sites** | 1 (referent) | 1.01 (0.95–1.07) | 0.98 (0.94–1.02) | 0.85 (0.78–0.91) |
| **Prostate** | 1 (referent) | 1.06 (0.95–1.18) | 1.00 (0.92–1.09) | 1.05 (0.93–1.20) |
| **Melanoma** | 1 (referent) | 1.16 (0.99–1.37) | 1.12 (0.99–1.28) | 1.03 (0.83–1.29) |
| **Non-Hodgkins** | 1 (referent) | 1.07 (0.81–1.42) | 0.90 (0.71–1.13) | 0.91 (0.61–1.37) |
| **Myeloma** | 1 (referent) | 0.84 (0.55–1.29) | 0.81 (0.57–1.14) | 0.42 (0.18–0.95) |
| **Leukaemia** | 1 (referent) | 1.08 (0.77–1.50) | 0.94 (0.72–1.22) | 0.76 (0.46–1.26) |
| **Lung** | 1 (referent) | 0.76 (0.61–0.94) | 0.84 (0.72–0.99) | 0.49 (0.34–0.71) |
| **Colon** | 1 (referent) | 1.08 (0.90–1.29) | 0.88 (0.76–1.02) | 0.71 (0.52–0.98) |
| **Bladder** | 1 (referent) | 0.89 (0.60–1.33) | 0.91 (0.67–1.23) | 0.51 (0.23–1.12) |
| **Kidney** | 1 (referent) | 0.96(0.67–1.39) | 1.04 (0.79–1.37) | 0.78 (0.48–1.27) |

Abbreviations:

[1] MoP = Metal-on-polyethylene,

[2] CoP = Ceramic-on-polyethylene,

[3] MoM = Metal-on-Metal,

[4] CoC = Ceramic-on-Ceramic, HR = Hazard ratio, CI = Confidence interval.

For csTHA procedures there was no significant difference in all-cause cancer rates for MoM bearing surfaces (Hazard Ratio (HR) 1.01, 95% CI 0.95–1.07) or CoC (HR 0.98, 95% CI 0.94–1.02) compared to MoP and CoP bearing surfaces Table 3. For rsTHA procedures there was a significantly lower risk of all-cause cancer compared to MOP and COP csTHA procedures. No significant differences in site-specific cancers between bearing surfaces were observed except for lung cancer which was found to have a lower risk in MoM (HR 0.76, 95% CI 0.61–0.94) and CoC (HR 0.84, 95% CI 0.72–0.99) compared to the comparison cohort. For rsTHA there was a lower risk of site-specific cancers including myeloma (HR 0.42, 95% CI 0.18–0.95), lung (HR 0.49, 95% CI 0.34–0.71) and colon (HR 0.71, 95% CI 0.52–0.98) cancer. Results for males and females were similar Table 4.

## Discussion

Patients with csTHA had an overall 24% higher risk of all-cause cancer than the Australian population with a similar age and gender profile. While many of the site-specific cancers showed elevated risk compared to standard Australian rates, no differences in risk was found between the bearing surfaces except for lung cancer with ceramic-on-ceramic procedures and metal-on-metal procedures compared to MOP and COP procedures. For rsTHA procedures there was a 42% higher risk of all-cause cancer compared to the Australia population, however, only prostate and melanoma cancers showed significantly increased risk. Resurfacing procedures had a lower risk of myeloma, lung and colon cancers compared to csTHA MoP and CoC procedures.

Our results showing a higher standardised incidence of cancer following total hip arthroplasty are similar to others. A Scottish study of total hip arthroplasty [38] found a significantly increased cancer incidence compared to the general population (SIR 1.05, 95% CI 1.04–1.07). However, in contrast to our results, studies in Finland [29, 30] and Sweden [25] have found no increased risk of all site cancer incidence. Similar to our study, previous studies have identified an increased risk of prostate cancer, skin melanoma and hematopoietic cancers among THA patients [16–22, 38].

**Table 4. Risk of cancer incidence by gender for all-cause and site-specific cancers, by bearing surface.**

| | Males | | Females | |
|---|---|---|---|---|
| | SIR (95% CI) | HR (95% CI) | SIR (95% CI) | HR (95% CI) |
| **All Cause** | | | | |
| **MoP CoP** | 1.26 (1.23–1.29) | 1.0 | 1.18 (1.14–1.21) | 1.0 |
| **MoM** | 1.38 (1.29–1.46) | 1.01 (0.95–1.09) | 1.22 (1.12–1.33) | 1.00 (0.91–1.10) |
| **CoC** | 1.36 (1.29–1.42) | 0.98 (0.93–1.04) | 1.20 (1.13–1.27) | 0.97 (0.91–1.04) |
| **Resurfacing** | 1.45 (1.34–1.56) | 0.85 (0.78–0.93) | 1.27 (1.07–1.48) | 0.91 (0.76–1.08) |
| **Leukaemia** | | | | |
| **MoP CoP** | 1.13 (0.95–1.31) | 1.0 | 1.26 (1.06–1.46) | 1.0 |
| **MoM** | 1.57 (1.01–2.14) | 1.32 (0.89–1.96) | 1.08 (0.44–1.72) | 0.74 (0.39–1.39) |
| **CoC** | 1.27 (0.89–1.64) | 1.06 (0.76–1.49) | 1.13 (0.69–1.58) | 0.77 (0.50–1.19) |
| **Resurfacing** | 1.24 (0.61–1.87) | 0.87 (0.49–1.53) | 1.38 (-0.18–2.94) | 0.65 (0.22–1.93) |
| **Melanoma** | | | | |
| **MoP CoP** | 1.56 (1.44–1.69) | 1.0 | 1.35 (1.22–1.48) | 1.0 |
| **MoM** | 1.90 (1.56–2.24) | 1.22 (1.00–1.48) | 1.42 (1.01–1.82) | 1.06 (0.78–1.43) |
| **CoC** | 1.80 (1.55–2.04) | 1.13 (0.96–1.33) | 1.47 (1.19–1.74) | 1.09 (0.87–1.36) |
| **Resurfacing** | 1.87 (1.50–2.25) | 1.15 (0.90–1.46) | 1.08 (0.49–1.66) | 0.74 (0.41–1.31) |

Abbreviations: MoP = Metal-on-polyethylene, CoP = Ceramic-on-polyethylene, MoM = Metal-on-Metal, CoC = Ceramic-on-Ceramic, HR = Hazard ratio,

SIR = Standardized incidence ratios, CI = Confidence interval, SIR = Standardized incidence ratios, CI = Confidence interval.

Overall, we found no difference in cancer risk with csTHA MoM compared to other bearing surfaces. This result is similar to UK studies, which also found no difference in overall risk of cancer with MoM compared to other bearing surfaces [26, 27, 32]. Other studies from Nordic countries of the association between MoM and cancer risk found no increased risk with MoM [16]. However, due to differences in the healthcare system and the low uptake of MoM prostheses in those countries, the Nordic studies may have limited power to examine MoM prostheses. Only one Finnish study compared MoM and MoP hip prostheses [22] finding no difference in all cause cancer incidence between the procedures. One study in Slovenia found that the risk for all cancers combined as well as for prostate and skin cancer was higher with MoM compared to non-MoM procedures (RR = 1.56; 2.02 and 1.92, respectively) [28].

Our study identified some variation in cancer risk by gender however results by gender were similar to the overall population. Similar findings were also observed in a UK study [26] which, in general, found that cause-specific cancer risks were greater with MoM compared to other bearing surfaces in males however results were not significant. For example, when we restricted our population to males there was a marginally significant increased risk of melanoma with MOM compared to the comparison (MoP or CoP) procedures (HR: 1.22, 95% CI 1.00–1.48).

The strength of our study is our long-term follow-up. Median follow-up was 5 years, with 25% of the population having 8 or more years of follow-up. Latency is an important consideration in determining the cause of cancer and is required from the initial exposure to a carcinogenic stimulus to a clinically diagnosed cancer. While there is uncertainty as to how quickly a cancer can be attributed to the prosthesis, it has been suggested that early excess cancer risk during follow-up may be due to detection bias or disease-related aetiological factors. To address this concern we excluded cancers diagnosed in the first year after joint replacement. Although most of the previous epidemiological studies included analyses of latency and duration of follow-up, the data are conflicting. A major issue is the reduced power when longer

latency periods are examined. In a meta-analysis of five Nordic cohorts of total hip arthroplasty, the combined data showed a modest but significantly positive trend for increased cancer risk with increased latency period [16]. Another strength of our study is our use of the Australian Cancer Registry for identification of cancer diagnosis. As cancer is a notifiable disease in Australia data are recorded for all new cases of cancer by each jurisdiction in Australia and are then collated, cleaned and standardised by the AIHW.

One of the weaknesses of this study is the lag in cancer data available from the Australian Cancer Registry at the time this study was conducted. Only cancers diagnosed to end of 2013 were available for analysis. While our study is the longest published to date, for many of the rare cancers only small numbers of incident cases were observed and longer term follow-up will be required. Secondly, the use of MoM hips (both large and small heads) has declined significantly over time and MoM procedures with large heads are no longer performed. Lastly, our use of SIR limited the effects of some confounding variables, age gender and calendar year, however it is possible that patients with total hip arthroplasties are generally healthier than the general population of similar age and this would likely bias our results towards a null effect [39]. On the other hand, some studies have suggested that cancer rates in joint replacement populations may be increased due to surveillance bias. It is unlikely that these biases will be differential between the MoM and other bearing surface group and therefore unlikely to have impacted on our analysis between these groups.

## Conclusions

Similar to other large cohort studies we have identified that cancer burden is generally higher in the cohort of patients receiving total hip arthroplasty procedures. However, this increased risk was not differentially higher for patients with MoM bearings compared to other bearing surfaces. Longer-term follow-up studies will be required to confirm these results for cancers with potentially longer latency periods.

## Supporting information

**S1 Table. ICD codes included in analysis.** Abbreviation: ICD-10 = International Classification of Diseases 10th Revision.
(DOCX)

## Acknowledgments

Data provision and linkage services.

We thank Michelle Lorimer, lead statistician with the AOANJRR for her assistance in providing the required AOANJRR data for linkage and analysis.

We thank the cancer registries across all Australian states and territories and the Australian Institute of Health & Welfare for access to the national Australian Cancer Database (ACD). We acknowledge the services of the Australian Institute of Health & Welfare Data Integration Services Centre for facilitating provision and linkage of the ACD with the AOANJRR.

## Author Contributions

**Conceptualization:** Nicole L. Pratt, Flavia M. Cicuttini, Yuanyuan Wang, Stephen E. Graves.

**Data curation:** Nicole L. Pratt.

**Formal analysis:** Nicole L. Pratt.

**Funding acquisition:** Nicole L. Pratt, Flavia M. Cicuttini, Yuanyuan Wang, Stephen E. Graves.

**Methodology:** Nicole L. Pratt, Flavia M. Cicuttini, Yuanyuan Wang, Stephen E. Graves.

**Project administration:** Nicole L. Pratt, Yuanyuan Wang.

**Software:** Nicole L. Pratt.

**Supervision:** Flavia M. Cicuttini.

**Visualization:** Nicole L. Pratt.

**Writing – original draft:** Nicole L. Pratt.

**Writing – review & editing:** Nicole L. Pratt, Flavia M. Cicuttini, Yuanyuan Wang, Stephen E. Graves.

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
