## [Decision Letter · Decision Letter 0]

22 Aug 2022

PONE-D-22-16723No increased risk of cancer after total hip arthroplasty associated with metal-on-metal hip arthroplasty: a nationwide linked registry cohort analysis of 167,837 patientsPLOS ONE

Dear Prof. Nicole Pratt,

Thank you for submitting your manuscript to PLOS ONE. After careful consideration, we feel that it has merit but does not fully meet PLOS ONE’s publication criteria as it currently stands. Therefore, we invite you to submit a revised version of the manuscript that addresses the points raised during the review process.The study is of interest and add knowledge to the the topic. However, there are some flaws in the methodology and the title should be more precise to match the results presented.

We look forward to receiving your revised manuscript.

Kind regards,

Arianna Di Napoli, M.D. Ph.D.

Academic Editor

PLOS ONE

Journal Requirements:

“We acknowledge seed funding from the Australian Orthopaedic Association and funding from the National Health and Medical Research Council (GNT1148106). “

“The following authors (YW, SEG, FMC & NLP) received funding from the Australian Orthopaedic Association Research Foundation (https://aoa.org.au/research/research-foundationM/research-foundation) for the work. Authors SEG and NLP also received funding from the Australian National Health & Medical Research Council Project Grant APP1148106 which supported the work (https://www.nhmrc.gov.au/funding/find-funding/project-grants). The funders played no role in the study design, data collection and analysis, decision to publish, or preparation of the manuscript.”

Reviewers' comments:

Reviewer's Responses to Questions

**Comments to the Author**

1. Is the manuscript technically sound, and do the data support the conclusions?

Reviewer #1: Yes

Reviewer #2: Yes

2. Has the statistical analysis been performed appropriately and rigorously? 

Reviewer #1: Yes

Reviewer #2: Yes

3. Have the authors made all data underlying the findings in their manuscript fully available?

Reviewer #1: No

Reviewer #2: Yes

4. Is the manuscript presented in an intelligible fashion and written in standard English?

Reviewer #1: Yes

Reviewer #2: Yes

5. Review Comments to the Author

Reviewer #1: Despite the large body of literature on this theme, the scientific work described in this paper is very interesting showing data on a region not yet investigated. The design and the analysis were well conducted. However, in my opinion the title seems a little misleading: in the abstract the authors said "no excess risk was observed for MoM or CoC procedures compared to other bearing surfaces". The lack of risk is related to the other implants, meanwhile in the title it seems the MoMs did not show a risk by themselves, respect to the general population. By this I suggest the authors to update the title.

I also suggest to add some few rows about the percentage of successful linkages because a low quota of linkages may have biased the study.

Minors to be updated:

Row (35) 156 , 516 population observed.

Row (161) 165, 516 procedures

Reviewer #2: Paper Title:

No increased risk of cancer after total hip arthroplasty associated with metal-on-metal hip arthroplasty: a nationwide linked registry cohort analysis of 167,837 patients

Reviewer comments

This observational retrospective study aims to evaluate the risk of all-cause and site-specific cancer in patients undergoing total hip arthroplasty and, in particular, when using Metal-on-Metal bearing surfaces. To this end data from the Australian Orthopaedic Association National Joint Replacement Registry (AOANJRR) have been linked to data collected by the Australasian Association of Cancer Registries.

This is in an interesting study and the findings presented may add knowledge on the topic. Anyway, some major and minor revisions are required to make it eligible for publication.

Major revisions:

Title:

- As the findings presented in the manuscript highlight no increased risk of cancer associated to MoM procedures as compared to other prosthesis (MoP, CoP), and not to the general population, should the Authors consider rephrasing the title?

Methods section:

- Page 5, Line 90: Authors state that “the AOANJRR has recorded all hip and knee joint replacement procedures from 1 September 1999, becoming completely national by mid-2002.” Did Authors consider not to include in the analyses the joint replacement procedures in this time window to prevent any potential selection bias?

- When selecting patients for the study, it would recommend to better explain how Authors were able to assess that the procedures recorded in the AOANJRR registry were related only to patients with osteoarthritis and not with other pathologies that may require hip replacement.

Minor revisions:

Methods section:

- Page 6, line 109: the acronyms “AACR” and “AIHW” are for?

- Page 6, line 116: the acronym “ACD” is for?

- I would recommend deleting the subsection title for “Australian orthopaedic association national joint replacement registry” and “Australian cancer database”. Consider to remove “Data linkage” as well, while I would suggest to change “Participants” into “Study population”.

- Page 7, Statistical analysis: What about mentioning the descriptive statistics presented in Table 1?

- Page 7, Line 135: Please refer to the calculation of 95% Confidence Intervals also.

Results section:

- Page 9, Line 161: Check the number of csTHA procedures (156.516 instead of 165.516?)

- S1 Table: Insert the meaning of the acronyms.

- Page 9, Line 169: a reference symbol to the note at the bottom is not present within the table

- Table 3: “SIR” and “CI” is for?

-Table 4: “HR” and “SIR” stand for?

6. PLOS authors have the option to publish the peer review history of their article (what does this mean?). If published, this will include your full peer review and any attached files.

Reviewer #1: No

Reviewer #2: No

---

## [Author Response · Author response to Decision Letter 0]

24 Oct 2022

Dear Dr Di Napoli,

Thank-you for your recent email (23 August 2022) requesting further information on our manuscript submission PONE-D-22-16723: “No increased risk of cancer after total hip arthroplasty associated with metal-on-metal hip arthroplasty: a nationwide linked registry cohort analysis of 167,837 patients”.

As per the advice received in the communication, we provide the following responses to the Reviewer comments and request to address additional requirements:

REVIEWER COMMENTS

Reviewer #1: 

1. Despite the large body of literature on this theme, the scientific work described in this paper is very interesting showing data on a region not yet investigated. The design and the analysis were well conducted. However, in my opinion the title seems a little misleading: in the abstract the authors said "no excess risk was observed for MoM or CoC procedures compared to other bearing surfaces". The lack of risk is related to the other implants, meanwhile in the title it seems the MoMs did not show a risk by themselves, respect to the general population. By this I suggest the authors to update the title.

AUTHOR RESPONSE: We have amended the title to read: “No increased risk of cancer associated with metal-on-metal or ceramic-on-ceramic procedures compared to other bearing surfaces in patients with total hip arthroplasty: a nationwide linked registry cohort analysis of 167,837 patients”.

2. I also suggest to add some few rows about the percentage of successful linkages because a low quota of linkages may have biased the study.

AUTHOR RESPONSE: The following text has been added to the Methods section (lines 120-123), “Records were successfully matched to the Medicare file for 97.8 % of patients with an AOANJRR-recorded hip arthroplasty procedure, with 20.1% of the Medicare-AOANJRR matched patients linked to the Australian Cancer Database.”

3. Minors to be updated:

Row (35) 156,516 population observed.

Row (161) 165,516 procedures

AUTHOR RESPONSE: Line 161 (now 162) has been corrected and now reads,”There were 156,516 csTHA patients included with median age 68 years and 5 years follow-up and 11,321 rsTHA patients with median age 53 years and 8 years follow-up.” 

Reviewer #2: 

Reviewer comments

This observational retrospective study aims to evaluate the risk of all-cause and site-specific cancer in patients undergoing total hip arthroplasty and, in particular, when using Metal-on-Metal bearing surfaces. To this end data from the Australian Orthopaedic Association National Joint Replacement Registry (AOANJRR) have been linked to data collected by the Australasian Association of Cancer Registries.

This is in an interesting study and the findings presented may add knowledge on the topic. Anyway, some major and minor revisions are required to make it eligible for publication.

Major revisions:

Title: 

4. As the findings presented in the manuscript highlight no increased risk of cancer associated to MoM procedures as compared to other prosthesis (MoP, CoP), and not to the general population, should the Authors consider rephrasing the title?

AUTHOR RESPONSE: As per our response to Reviewer 1’s comment we have amended the title to read: “No increased risk of cancer associated with metal-on-metal or ceramic-on-ceramic procedures compared to other bearing surfaces in patients with total hip arthroplasty: a nationwide linked registry cohort analysis of 167,837 patients”.

Methods section: 

5. Page 5, Line 90: Authors state that “the AOANJRR has recorded all hip and knee joint replacement procedures from 1 September 1999, becoming completely national by mid-2002.” Did Authors consider not to include in the analyses the joint replacement procedures in this time window to prevent any potential selection bias?

AUTHOR RESPONSE: The implementation of joint replacement data collection in Australia occurred in a staged approach on a state-by-state basis. The majority of Australia’s jurisdictions (representing 66% of the Australian population) were contributing data by September 2000 with the final state commencing in June 2001. 

While the authors considered restricting the joint replacement procedures for analysis from mid-2002 onwards, however the impact of selection bias was considered minimal as the decision to use a particular bearing surfaces is unlikely driven by geographical location. 

No changes to the additional manuscript have been made. 

6. When selecting patients for the study, it would recommend to better explain how Authors were able to assess that the procedures recorded in the AOANJRR registry were related only to patients with osteoarthritis and not with other pathologies that may require hip replacement.

AUTHOR RESPONSE: Information collected by the Registry at time of surgery includes the diagnosis underlying the reason for joint replacement surgery. This diagnosis is recorded by the surgical team at the time of surgery.

We have amended the manuscript text (Methods, lines 92-93) to read, “The AOANJRR collects a defined minimum dataset at time of surgery that comprises patient characteristics including age and sex, surgical indication (diagnosis reason for surgery e.g. osteoarthritis, tumour, fractured neck of femur)…”

Minor revisions:

Methods section:

7. Page 6, line 109: the acronyms “AACR” and “AIHW” are for?

AUTHOR RESPONSE: The AACR acronym has been added with the first use of the full phrase (Methods, line 88) while specified in full on line 108. 

The AIHW acronym has been replaced with its full reference (Methods, lines 111-112). 

8. Page 6, line 116: the acronym “ACD” is for?

AUTHOR RESPONSE: The acronym ACD has now been specified in full at first use of the phrase, Australian Cancer Database (Methods, lines 111-112). 

9. I would recommend deleting the subsection title for “Australian orthopaedic association national joint replacement registry” and “Australian cancer database”. Consider to remove “Data linkage” as well, while I would suggest to change “Participants” into “Study population”.

AUTHOR RESPONSE: The three subsection titles appearing in the Methods section have now been deleted:

• Australian Orthopaedic Association National Joint Replacement Registry, 

• Australian Cancer Database and,

• Data Linkage.

As recommended, the Participants sub-title has been replaced by the term Study population (Methods Section, line 124).

10. Page 7, Statistical analysis: What about mentioning the descriptive statistics presented in Table 1?

AUTHOR RESPONSE: A sentence has been added to the Methods, Statistical analysis subsection (lines 143-145), “Descriptive statistics were used to summarise the cohort demographics and follow-up time for the two classes of hip arthroplasty, csTHA and rsTHA and three bearing surfaces (MoM, CoC, MoP/CoC combined).” 

11. Page 7, Line 135: Please refer to the calculation of 95% Confidence Intervals also.

AUTHOR RESPONSE: Statement referring to calculation of 95% Confidence Intervals now added (Methods, Statistical analysis subsection line 146), “Standardized incidence ratios (SIRs) with 95% confidence intervals (CIs) were calculated for all cancer types and for site-specific cancer (Table in S1 Table).” 

Results section:

12. Page 9, Line 161: Check the number of csTHA procedures (156,516 instead of 165,516?)

AUTHOR RESPONSE: The typographical error has been corrected and now reads 156,516 csTHA patients (Results section, line 172).

13. S1 Table: Insert the meaning of the acronyms.

AUTHOR RESPONSE: In Supplementary Table 1, acronym ICD-10 has been specified in full as the International Classification of Diseases 10th Revision. The description for each ICD-10 code used in the analysis has also been specified in full. 

14. Page 9, Line 169: a reference symbol to the note at the bottom is not present within the table

AUTHOR RESPONSE: Reference symbol, * now added to Table 1 Bearing surface heading (column 1, row 1). 

15. Table 3: “SIR” and “CI” is for? 

AUTHOR RESPONSE: The acronyms SIR and CI representing Standardized incidence ratio and Confidence interval, respectively have been specified in full in Tables 2 and 4. Note acronym SIR does not appear in Table 3.

16. Table 4: “HR” and “SIR” stand for?

AUTHOR RESPONSE: The acronyms HR and SIR representing Hazard ratio and Standardized incidence ratio, respectively, have been specified in full in Table 4 (and Table 3).

ADDITIONAL REQUIREMENTS

• PLOS ONE Style Requirements

We confirm the manuscript meets PLOS ONE’s style and file naming requirements as per the PLOS ONE style templates.

• Funding Statement 

As requested, we confirm removal of the following text relating to funding information from the Acknowledgements Section of our manuscript:

“We acknowledge seed funding from the Australian Orthopaedic Association and funding from the National Health and Medical Research Council (GNT1148106). 

YW is the recipient of National Health and Medical Research Council Translating Research into Practice Fellowship (APP1168185).”

Our Funding Statement has been updated and now reads: 

“The following authors (YW, SEG, FMC & NLP) received seed funding from the Australian Orthopaedic Association Research Foundation (https://aoa.org.au/research/research-foundationM/research-foundation) for the work. Authors SEG and NLP also received funding from the Australian National Health & Medical Research Council Project Grant APP1148106 which supported the work (https://www.nhmrc.gov.au/funding/find-funding/project-grants).

YW is the recipient of National Health and Medical Research Council Translating Research into Practice Fellowship APP1168185 (https://www.nhmrc.gov.au/sites/default/files/documents/attachments/Outcomes/TRIP-Fellowship.pdf).

The funders played no role in the study design, data collection and analysis, decision to publish, or preparation of the manuscript.”

• Data accessibility

We confirm removal of the phrase ‘data not shown’ relating to results of sensitivity analyses.

We thank you for your ongoing consideration of our manuscript.

---

## [Decision Letter · Decision Letter 1]

14 Nov 2022

No increased risk of cancer after total hip arthroplasty associated with metal-on-metal hip arthroplasty: a nationwide linked registry cohort analysis of 167,837 patients

PONE-D-22-16723R1

Dear Dr. Pratt,

We’re pleased to inform you that your manuscript has been judged scientifically suitable for publication and will be formally accepted for publication once it meets all outstanding technical requirements.

Kind regards,

Arianna Di Napoli, M.D. Ph.D.

Academic Editor

PLOS ONE

Additional Editor Comments (optional):

Reviewers' comments:

Reviewer's Responses to Questions

**Comments to the Author**

1. If the authors have adequately addressed your comments raised in a previous round of review and you feel that this manuscript is now acceptable for publication, you may indicate that here to bypass the “Comments to the Author” section, enter your conflict of interest statement in the “Confidential to Editor” section, and submit your "Accept" recommendation.

Reviewer #1: All comments have been addressed

Reviewer #2: All comments have been addressed

2. Is the manuscript technically sound, and do the data support the conclusions?

Reviewer #1: Yes

Reviewer #2: Yes

3. Has the statistical analysis been performed appropriately and rigorously? 

Reviewer #1: Yes

Reviewer #2: Yes

4. Have the authors made all data underlying the findings in their manuscript fully available?

Reviewer #1: Yes

Reviewer #2: Yes

5. Is the manuscript presented in an intelligible fashion and written in standard English?

Reviewer #1: Yes

Reviewer #2: Yes

6. Review Comments to the Author

Reviewer #1: The authors have successfully updated the manuscript as suggested, the paper is now ready for publication

Reviewer #2: (No Response)

7. PLOS authors have the option to publish the peer review history of their article (what does this mean?). If published, this will include your full peer review and any attached files.

Reviewer #1: No

Reviewer #2: **Yes: **Walter Mazzucco

---

## [Editor Report · Acceptance letter]

18 Nov 2022

PONE-D-22-16723R1 

No increased risk of cancer associated with metal-on-metal or ceramic-on-ceramic procedures compared to other bearing surfaces in patients with total hip arthroplasty: a nationwide linked registry cohort analysis of 167,837 patients. 

Dear Dr. Pratt:

I'm pleased to inform you that your manuscript has been deemed suitable for publication in PLOS ONE. Congratulations! Your manuscript is now with our production department. 

Kind regards, 

on behalf of

Dr. Arianna Di Napoli 

Academic Editor

PLOS ONE